Identification of a genome-specific repetitive element in the Gossypium D genome

Lu Hejun 1 2
Cui Xinglei 2
Zhao Yanyan 2
Magwanga Richard Odongo 2 3
Li Pengcheng 2
Cai Xiaoyan 2
Zhou Zhongli 2
Wang Xingxing 2
Liu Yuling 4
Xu Yanchao 2
Hou Yuqing 2
Peng Renhai 4
Wang Kunbo wkbcri@163.com 2 5
Liu Fang liufcri@163.com 2
1 Gembloux Agro-Bio Tech, University of Liège , Gembloux , Namur , Belgium
2 Research Base of Tarium University, State Key Laboratory of Cotton Biology, Institute of Cotton Research of Chinese Academy of Agricultural Science , Anyang , Henan , China
3 School of Biological and Physical Sciences (SBPS), Jaramogi Oginga Odinga University of Science and Technology (JOOUST), Bondo-Kenya , Bondo , Kenya
4 Anyang Institute of Technology , Anyang , Henan , China
5 Tarium University , Alar , Xinjiang , China
Lazo Gerard
Electronic publication date: 2020 Jan 3
Publication date: 2020
Volume: 8
Electronic Location ID: e8344
Received 2019 Jun 14; Accepted 2019 Dec 4
Copyright: ©2020 Lu et al.
Copyright year: 2020
Copyright holder: Lu et al.
License: This is an open access article distributed under the terms of the Creative Commons Attribution License, which permits unrestricted use, distribution, reproduction and adaptation in any medium and for any purpose provided that it is properly attributed. For attribution, the original author(s), title, publication source (PeerJ) and either DOI or URL of the article must be cited.
License URL: https://creativecommons.org/licenses/by/4.0/

Keywords: Gossypium, Repetitive element, D genome, Fluorescence in situ hybridization (FISH), Genome-specific, Evolution

Funding: The National Key Research and Development Plan of China 2016YFD0100306 2016YFD0100203 The Natural Science Foundation of China 31530053 31671745 This research was supported by the National Key Research and Development Plan of China (grants 2016YFD0100306 and 2016YFD0100203) and The Natural Science Foundation of China (grants 31530053 and 31671745). The funders had no role in study design, data collection and analysis, decision to publish, or preparation of the manuscript.

==============================
The activity of genome-specific repetitive sequences is the main cause of genome variation between Gossypium A and D genomes. Through comparative analysis of the two genomes, we retrieved a repetitive element termed ICRd motif, which appears frequently in the diploid Gossypium raimondii (D5) genome but rarely in the diploid Gossypium arboreum (A2) genome. We further explored the existence of the ICRd motif in chromosomes of G. raimondii, G. arboreum, and two tetraploid (AADD) cotton species, Gossypium hirsutum and Gossypium barbadense, by fluorescence in situ hybridization (FISH), and observed that the ICRd motif exists in the D5 and D-subgenomes but not in the A2 and A-subgenomes. The ICRd motif comprises two components, a variable tandem repeat (TR) region and a conservative sequence (CS). The two constituents each have hundreds of repeats that evenly distribute across 13 chromosomes of the D5genome. The ICRd motif (and its repeats) was revealed as the common conservative region harbored by ancient Long Terminal Repeat Retrotransposons. Identification and investigation of the ICRd motif promotes the study of A and D genome differences, facilitates research on Gossypium genome evolution, and provides assistance to subgenome identification and genome assembling.

Introduction

Repetitive DNA sequences are common in eukaryotic genomes, and account for a large fraction of the host genome (Ibarra-Laclette et al., 2013). Their amount is highly correlated with the size of the host genome (Feschotte, 2008). Repetitive DNA is divided into two major groups based on their structures: tandem repeats and interspersed repeats (Jurka et al., 2005). Tandem repeats are known as simple sequence repeat (SSR), and include micro-satellites, mini-satellites, and satellites (Jeffreys, Neumann & Wilson, 1990). Interspersed repeats are also referred to as transposable elements (TEs) or transposons.

After the first TE of Ac/Ds was reported in maize (McClintock, 1950; Brink & Williams, 1973; Goldschmidt, 2002), further TEs have been identified in many eukaryotic species (Munoz-Lopez & Garcia-Perez, 2010). There are thousands of different TE families in plants, which display extreme diversity (Sanmiguel & Bennetzen, 1998; Bennetzen, 2005; Morgante, 2006). Finnegan first proposed a TE classification system, which includes two classes based on their transposition mechanisms, viz., those mediated by RNA (Retrotransposons) and those by DNA (DNA transposons) (Bowen & Jordan, 2002; Wessler, 2006; Arkhipova, 2018). Wicker unified TEs nomenclature and classification by applying mechanistic and enzymatic criteria (Wicker et al., 2007). TEs play important roles in the genome through diverse ways, such as variation in intron size (Deutsch & Long, 1999; Zhang et al., 2011; Koonin, Csuros & Rogozin, 2013), segmental duplication (Del Pozo & Ramirez-Parra, 2015), transfer of organelle DNA to the nucleus (Adams & Palmer, 2003), expansion/contraction of tandem repeats, and illegitimate recombination (Finnegan, 1989; Koike, Nakai & Takagi, 2002). Long Terminal Repeat Retrotransposons (LTR-TEs), which are usually scattered throughout genomes, are the most abundant TE type and can cause genome expansion over a short evolutionary period particularly in plants (Piegu et al., 2006). The investigation of genome-specific TE is an efficient approach to studying species formation and genome evolution (Dong et al., 2018).

Gossypium, a genus of flowering plants from which cotton is harvested, diverged from the common ancestor with Theobroma cacao approximately 33.7 million years ago (MYA) (Wang et al., 2012). Gossypium comprises eight diploid (2n = 2x = 26) genomic groups: A, B, C, D, E, F, G, K, and one allotetraploid (2n = 4x = 52) genomic group: AD (Wang, Wendel & Hua, 2018). Gossypium species are good subjects for research on polyploidization, genomic organization and genome-size variation because of their high genome diversity: from the smallest New World D genome with an average of 885 Mb to the Australian K-genome with an average of 2,576 Mb (Hendrix & Stewart, 2005). The accumulation of different lineage-specific TEs was thought to be responsible for the variation in genome size in Gossypium genomic groups (Hawkins et al., 2006; Lu et al., 2018b). Of the eight genomic groups, the A and D groups are the main ones investigated in cotton genomics research (Du et al., 2018). Gossypium hirsutum, the major cultivated cotton species, is known to have originated from the progenitors of G. arboreum (A2) and G. raimondii (D5) (Paterson et al., 2012). The key phenotype difference between G. arboreum and G. raimondii is the production of spinnable fibers in the former but not the latter. In terms of the genomics, the former has a genome size of 1,746 Mb/1C, which is about two times that of the latter (885 Mb/1C) (Hendrix & Stewart, 2005). Genome sequencing showed that the difference in the numbers of protein-coding genes between the A (41,330) and D (37,505) genomes is not obvious, while the lineage-specific TE content is the main reason for the size gap between the A and D genome (Li et al., 2015; Du et al., 2018). Moreover, Wang, Huang & Zhu (2016) suggested that the transposable elements play an important role during cotton genome evolution and fiber cell development. Thus, research on the lineage-specific repetitive sequences between A and D genomes is a meaningful path to investigate speciation dynamics.

Fluorescence in situ hybridization (FISH) is a versatile tool to visualize the distribution of certain DNA sequences in chromosomes and plays a vital role in cytogenetic research. In tetraploid cotton, FISH has played a key role in cytological experiments that have contributed to the understanding the allotetraploid event. FISH with DNA segments harboring dispersed repeats has identified genome-specific repeats between the A and D genome, and showed that some A genome repeats appear to have spread to the D genome (Hanson et al., 1998; Zhao et al., 1998a; Zhao et al., 1998b). Although the repetitive DNA fragments are more common in the A than in the D genome, one tandem repeat family (B77) has been well-characterized from the D Chromosome (Zhao et al., 1998a; Zhao et al., 1998b). Recently, more repetitive sequences were observed with FISH in the cotton genome after construction of a cotton cytogenetic map Cui et al., 2015; Liu et al. 2016. Lu et al., 2018) suggested that CICR was an important contributor to the size gap between the A and D genome. The identification and localization of these repetitive sequences benefit genome assembly and facilitate understanding of the mechanism of genome evolution.

The D genomic group represents a diverse group of diploids that diverged from a branch of A, B, C, E, F, G, and K genomic groups about 5–10 MYA (Senchina et al., 2003). Although the D genome has the smallest size of all Gossypium species, this study has revealed the presence of a set of repeat elements with high proliferation, which is absent in the A genome. The discovery and characterization of these novel repetitive elements provides components for a repetitive sequences database and new insight into Gossypium evolution.

Materials and Methods

Plant materials

Cotton plants were obtained from the National Wild Cotton Nursery in Hainan Island, China, sponsored by the Institute of Cotton Research of Chinese Academy of Agricultural Sciences (ICR-CAAS). They were also conserved in the greenhouse at ICR-CAAS’ headquarters in Anyang City, Henan Province, China. The DNA and cells came from specimens listed in Table 1, which is based on the latest nomenclature of Gossypium species (Wang, Wendel & Hua, 2018).

Table 1 Plant materials used in this work, together with ploidy, studied genome, and specimen accession code.

Species	Ploidy	Genome	Accession	
G. arboreum	2x	A2	Shixiya-1	
G. raimondii	2x	D5	D5-07	
G. hirsutum	4x	(AD)1	CCRI-12	
G. barbadense	4x	(AD)2	Xinhai-7	

The repeat elements were characterized in the G. raimondii genome (Paterson et al., 2012), and compared to genomes in other Gossypium genomes, viz., G. arboreum (Li et al., 2014), G. hirsutum (AD)1 (BGI (Li et al., 2015), NBI (Zhang et al., 2015), HAU (Wang et al., 2019), ZJU (Hu et al., 2019)), G. barbadense (AD)2 (HAUv1 (Yuan et al., 2015), CAS (Liu et al., 2015), HAUv2 (Wang et al., 2019), and ZJU (Hu et al., 2019)). All genome data was downloaded from Cottongen (https://www.cottongen.org/), except the (AD)2-CAS which was obtained from GenBank under PRJNA251673.

Characterization of the repetitive element and bioinformatics analysis

BLASTN (v2.6.0) (Camacho et al., 2009) was used to identify repeat elements in the genomes of the plant material, and in the genomes stored in the databases. We used a threshold of greater than or equal to 80% matching ratio and an 80% similarity following the 80–80 rule suggested by Wicker et al. (2007). The tandem repeats (TRs) were identified with Tandem Repeats Finder (v4.09) (Benson, 1999). We used Perl script for batch extracting sequences from the genome (Doc S1). Sequence alignments were obtained from MUSCLE (v3.81) (Edgar, 2004). The Unipro UGENE (v1.31) was used to present the alignments and train consensus sequences (Okonechnikov et al., 2012). The inner enzyme annotation was obtained by online CD-search in NCBI (Marchler-Bauer et al., 2017). GIRI Repbase (Chen et al., 2007) were queried for annotation. RepeatMasker (v4.07) was used to annotate the insertions and estimate the proportion of repetitive sequences in genomes (http://www.repeatmasker.org).

Flanking LTRs of LTR-TEs were identified with LTRharvest (v1.5.8) (Ellinghaus, Kurtz & Willhoeft, 2008). Subsequently, Vmatch (v2.3.0) was used to cluster the LTRs (Kurtz, 2003). The divergence time of the LTR-TEs was estimated using the formula T = d∕2r, where r represents a substitution rate of 1.3 × 10−8 per site per year (Ma & Bennetzen, 2004), and d represents the distances of paired LTRs, which was calculated based on the Kimura two-parameter (Kimura, 1980). The insertions of the repeat elements were obtained based on the BLASTN result, and the LTR-TE and Coding-sequence (CDS) information was obtained from genome annotation (Paterson et al., 2012), which were illustrated by the ggplot2 R package (Wickham, 2016) with a sliding 500 kb window for LTR-TE and CDS. The synteny blocks of the homologous segments were shown by a Perl script (Doc S1) based on the BLASTN results.

Fluorescence in situ hybridization (FISH)

A probe was designed with the PCR product of the ICRd motif, which was obtained from the forward primer: TTCTATTTTATCCATCGTTATG, reverse: GGAGATAGGATTTGTTGCT; and followed the amplification procedure: firstly, 95 °C for 5 min of pre-degeneration; then, 30 cycles at 95 °C for 30 s, 52 °C for 30 s, and 72 °C for 2 min. The final extension was done at 72 °C for 6 min. Composition of the reaction mix used the following: gDNA (∼5 µg/ml), primers (∼0.8 µM), PCR Master Mix (Thermo), and H2O. The gDNA was extracted from the leaves of the cotton plants (Table 1). The probe was purified and labeled with digoxigenin-dUTP via nick translation, according to manufacturer’s instructions (Roche Diagnostics, USA). Mitotic chromosome preparation and FISH procedures were conducted using a modified protocol (Wang et al., 2001).

Figure 1 The structure of ICRd motif.

(A) The self-blast of the ICRd motif showed the inner repeats; (B) the structure of ICRd motif; (C) the basic TR unit; (D) the examples of the structure illustration of the LTR-TEs inserted with ICRd motif.

Results

One specific repetitive sequence in the Gossypium D5 genome

We performed BLAST to query all of the interspersed repetitive sequences of G. raimondii (Paterson et al., 2012) with the whole genome of G. arboreum (A2) (Li et al., 2014). One segment in the G. raimondii (D5) genome (Chr05: 50639971-50641791) was filtered out and recognized as D5 genome-specific. This sequence repeats frequently and is spread over 13 chromosomes of the D5 genome (Table S1), while it is absent from the A2 genome. Searches in Repbase (Chen et al., 2007) and NCBI found no related annotation and LTRharvest (Ellinghaus, Kurtz & Willhoeft, 2008) and a CD-search (Marchler-Bauer et al., 2017) revealed it is neither LTR nor a coding sequence.

Manual inspection revealed the structure of the genome-specific sequence as having two constituents: a tandem repeats array (referred as TR hereafter) composed of 133 bp basic units, and an unknown conservative sequence (referred as CS hereafter) (Fig. 1). Based on this feature, we identified 72 sequences in total from the D5 genome with RepeatMasker (Table S2), referred to here as the ICRd motif following our previous work (Lu et al., 2018b). Among the 72 ICRd motifs, the TRs are length-variable having 2–20 times of basic units (Fig. 2A), while the CSs are stable and have an average size ∼860 bp.

Figure 2 The content of the basic unit in the TRs.

(A) The basic unit content in the TRs involved in the ICRd motifs, displayed from small to large; (B) the number of ICRd TRs that harboring different unit content, the x-axis adopt the intervals of unit content for convenient exhibition.

To verify the genome specificity and chromosome distribution of the ICRd motif, we used the PCR product of the ICRd motif from G. raimondii to design the probe for FISH on the mitotic chromosomes of diploid A2 andD5, and tetraploid G. hirsutum ((AD)1) and G. barbadense ((AD)2). The probe generated bright signals covering all the chromosomes of the D5 and D-subgenome, but no signals on the A2 and A-subgenome (Fig. 3). These cytogenetic inspections were in accordance with the genomic comparative analysis and further revealed that the ICRd motif is a genome-specific and highly repetitive element in the D5 genome, as well as in the D-subgenome of tetraploid cotton.

Figure 3 The FISH images of ICRd. motif (red) hybridized to mitotic chromosomes of four species.

(A) G. arboreum (AA); (B) G. hirsutum (AADD); (C) G. barbadense (AADD); (D) G. raimondii (DD). Bar = 5 µm.

LTR-TEs inserted with the ICRd motif

We compared the insertion loci of 72 ICRd motifs with the whole genome repeats annotation (gff file) of the D5genome (Paterson et al., 2012) and found that each of the motifs is one-to-one harbored within the 72 LTR-TEs (Table S3), which meant the former is the inner part of the latter.

We extracted the 72 LTR-TEs sequences from the D5 genome and parsed their structure, which showed all sequences are incomplete, lacking either enzyme or flanking LTRs, the required elements for an intact LTR-TE (Wicker et al., 2007). A consensus accumulation histogram obtained from aligning all of these LTR-TEs together (Fig. S1) showed these TEs to have a vast sequence variation and a single conservative region representing the insertion region of the ICRd motif (Fig. 4). The ICRd motif appears to be more stable than other parts of the TEs along with degradation and evolution. This stability implies the importance of ICRd motif to the TEs.

Figure 4 The consensus accumulation histogram from the whole alignment of the 72 LTR-TEs.

The region marked with the black line is the ICRd motif region.

Of the 72 LTR-TEs, 25 were identified as having paired flanking LTRs, and were used to represent the classification and evolution of these TEs. The LTR cluster results showed that, except for two TEs having similar LTR regions, the other 23 TEs are totally different from each other, indicating that they do not belong to the same family based on the LTR similarity rules (Wicker et al., 2007). The estimated active date curve of these TEs—almost all prior to 10 MYA and peaking at ∼30 MYA (Fig. 5)—shows the peak is close to the time that G. raimondii and T. cacao diverged approximately 33.7 MYA (Wang et al., 2012), far earlier than the putative divergence time of the Gossypium A and D genomes (Wendel & Cronn, 2001). These results indicate that these LTR-TEs are ancient and potentially contributed to speciation of Gossypium.

Figure 5 The accumulation of putative active date of the LTR-TEs.

Abundant constituents of the ICRd motif in the D5 genome

To further analyze the genomic features of the ICRd motif, we separately investigated the content and distribution of its two constituents (TR and CS) in the D5 genome (Fig. 6A). In total 350 TR insertions were detected (Table S2). Insertions varied in length (due to the unit repeating at different times) between 2–21, but mainly 2–10 times the basic unit length (Fig. 2B). The longest TR insertion in D5 (D503: 25689303–25697234) was an extraordinary 61 units up to 8 kb; how it was formed is unknown. On the other hand, a total of 463 CSs were found (Table S2). Combining the analyses of the insertion loci of the two constituents, we found 72 TRs and 72 CSs constituting the ICRd motifs (Fig. 1).

Figure 6 The distribution of the ICRd motif and its constituent in the D5 genome.

(A) Insertions of the ICRd motif and its constituents in the D5 genomes; (B) (C) ICRd TR and TR-c chromosomal distribution, the expected (grey) and actual (white) distributions across all chromosomes are illustrated; in addition, the density per megabase is shown for each chromosome.

Further analysis showed that the TR and CS were evenly distributed on the chromosomes based on an χ2 test, with the number of insertions being proportional to the size of the chromosome [TR insertions, χ2 = 5.56 (df = 12, P > 0.9); CS insertions, χ2 = 9.08 (df = 12, P > 0.5)]. The even distributions meant that the CS and TR are possible ancient repetitive sequences that have evolved along with the chromosomes. Previous G. raimondii genome sequencing work reported that most TEs in G. raimondii are deletion derivatives lacking the domains that are typically necessary for transposition and that only 3% of LTR base pairs derived from full-length LTR-TEs (Paterson et al., 2012). We show that hundreds of constituents of the ICRd motif in D5 are potentially the fragments produced from the ancient LTR-TEs.

Disappearance of the ICRd motif from Gossypium

Aiming to observe the disappearance of the ICRd motif in the newly formed Gossypium A genome, we selected two homologous segments from the highly collinear Chromosome 1 of G. raimondii (D501) and G. arboreum (A201) (Li et al., 2014), respectively. The segment from Chromosome 1 of G. raimondii (D501) harbored one ICRd motif and its homologous segment from A201 was obtained from homologous SSR markers (Table S4). The illustration of the syntenic block of the two segments showed that the ICRd motif together with its host LTR-TE were lost on the A201 segment, while their up- and downstream conservative regions remained (Fig. 7). In the upstream, we observed two insertions rich in repeat sequences especially on the A201 segment (Table S4), which was potentially due to a recent TE expanding event happening in the A genome (Lu et al., 2018a). Thus, we observed that the ICRd motifs and host LTR-TEs were lost from the genome with the recent formation of the A genome (Wendel & Cronn, 2001; Wendel, Flagel & Adams, 2012), but remained in the D genome despite mass damage accumulation.

Figure 7 The colinearity of the two homologous segments.

Distributions of ICRd motifs in tetraploid cotton

Tetraploid cotton, G. hirsutum and G. barbadense, are the major cultivated fiber-producing cotton species. Research on the genome of these two species is an important approach to improving cotton yield and quality. However, due to the large number of homologous segments between A and D-subgenomes, the tetraploid cotton genome assemblage has been a great challenge to molecular geneticists (Bowers et al., 2003; Chen et al., 2007). Through high-throughput sequencing methods, two versions of the G. hirsutum genome assembly ((AD)1-BGI (Li et al., 2015), (AD)1-NBI (Zhang et al., 2015), and two G. barbadense versions (AD)2-HAU (Yuan et al., 2015) and (AD)2-CAS (Liu et al., 2015) were completed in 2015. With the advance of sequencing techniques, the tetraploid genome assemblies were improved in quality (Hu et al., 2019; Wang et al., 2019). However, to benefit research in the post-genome era, such as facilitating molecular breeding of cotton, suitable evaluation is needed to provide accurate reference data. Application of the lineage-specific repetitive element and the ICRd motifs are important tools in evaluating the quality of the genome assembly of tetraploid cotton.

To observe the assembling quality of the ICRd motif in tetraploid genomes, we queried it with BLAST in all published tetraploid cotton genomes, including four versions of G. hirsutum ((AD)1) and four versions of G. barbadense ((AD)2) (Table 2). In the case of (AD)1, the two recently published (Hu et al., 2019; Wang et al., 2019) versions and the previous NBI version were in agreement with our FISH inspection results, viz., that the ICRd motifs only generated the signals on the D-subgenome chromosomes (Fig. 3). However, the BGI version (Li et al., 2015) is inconsistent with the FISH results in that the ICRd motif was misassembled into the A-subgenome. For the (AD)2 assemblies, the two newly published (Hu et al., 2019; Wang et al., 2019) and CAS versions were better assembled than the HAUv1 version. The HAUv1 showed the number of matches in the chromosome-unassembled scaffolds, while the HAUv2 has improved quality (Table S5). This means that with advances in genome sequencing techniques, tetraploid genomes can be more precisely assembled though the existence of homologous segments from At and Dt.

Table 2 The distribution of ICRd motifs on different genome assemblies of tetraploid cotton.

Tetraploid	Version	Reference	ICRd motif	
G. hirsutum (AD)1	BGI	Li et al. (2015)	Dh01-Dh13; Ah02, Ah05, Ah07, Ah08	
NBI	Zhang et al. (2015)	Dh01-Dh13; None in A-sub	
HAU	Wang et al. (2019)	Dh01-Dh13; None in A-sub	
ZJU	Hu et al. (2019)	Dh01-Dh13; None in A-sub	
G. barbadense
(AD)2	CAS	Liu et al. (2015)	Db01-Db13; None in A-sub	
HAUv1	Yuan et al. (2015)	Db01, Db02, Db06-Db09, Db12; None in A-sub	
HAUv2	Wang et al. (2019)	Dh01-Dh13; None in A-sub	
ZJU	Hu et al. (2019)	Dh01-Dh13; None in A-sub	

Discussion

Identification of ICRd motif and Gossypium evolution

TEs have played an important function in Gossypium speciation and the accumulation of different genomic-specific TEs were thought to be responsible for genome-size variation in Gossypium (Hawkins et al., 2006). Through FISH inspection, some A genome-specific repetitive elements have been well identified and characterized (Liu et al., 2016), but similar work in the D genome have been rare; this may be because the genome-specific repetitive sequences in the A genome are much more numerous than in the D genome (Liu et al., 2018a). However, in the present study, starting with comparative genomic data, we have screened out one kind of specific sequence in the D genome, and subsequently, we have identified and characterized TEs.

The TEs harboring the ICRd motif showed an ancient active date of much earlier than 10 MYA, while the time of divergence of the A and D genomes from the common ancestor is estimated to have occurred 5–10 MYA (Grover et al., 2004). Thus the ICRd motifs have existed in the ancestor of A and D genome, while disappeared along with the formation of the A genome. Previous researchers have considered that the accumulation of lineage-specific TEs, which is thought to be responsible for the variation of Gossypium genomes (Hawkins et al., 2006), and the LTR-TE activities after 5 MYA mainly contributed to the two-fold size difference of the A and D genomes (Zhang et al., 2015). Based on our analysis, we presumed that as in the activity of new repetitive sequences the extinction of ancient repetitive sequences, such as the disappearance of the ICRd motif in the A genome, also contributed significantly to genome evolution. Through FISH, we observed that the ICRd motifs were only distributed in D-subgenome chromosomes, and the results were in agreement with a previous study which reported that the TE have proliferated in the progenitor genomes but were retained after allopolyploid formation in the D-subgenome (Zhang et al., 2015).

ICRd motif support cytogenetic markers for tetraploid cotton

The identification of the ICRd motif provides a new subgenome marker for the accurate assembling of tetraploid cotton (Chen et al., 2007). Chromosome identification is the foundation of plant genetics, evolution and genomics research (Saranga, 2007; Xie et al., 2012). Although many species have been sequenced, the rapid identification of the subgenome is still useful in applied research. FISH has been used as a reliable cytological technique for chromosome identification in many species (Wang, Guo & Zhang, 2007), but has only been used recently for the identification of cotton chromosomes (Gan et al., 2012). In the present study, the identified ICRd motifs can be used as a new cytological marker in Gossypium, especially in tetraploids. Further, the repetitive sequence probes are easier and more successfully detected than other probes. Several similar markers have been reported (Liu et al., 2016). The addition of these new cytological markers will enrich the marker database for chromosome identification and facilitate cotton genomic studies.

Eukaryotic genomes have a high proportion of TEs and these TEs make eukaryotic genome assembly much more difficult than simple genome assembly (Treangen & Salzberg, 2012). Many reported genome sequences have gaps because of the high proportion of TEs (Adams et al., 2000). Allopolyploid genomes are especially difficult to assemble homologous fragments from subgenomes (Chen et al., 2007). Incorrect assembling of the genomes leads to ambiguity in research which, in turn, produces biases and errors when interpreting results (Adams et al., 2000). The repetitive sequences analysis in this work were screened out from the whole genome comparison, we characterize the distribution feature on referenced genome assembly, moreover, FISH observation on chromosomes of somatic cell verified the lineage-specific feature. Combining FISH with genome-specific repeat segments is a convenient and practical approach to observe chromosome differences, in addition to assisting polyploid genome assembling, and evaluating assembling accuracy. With the progress of genome sequencing and assembling, genome assembly will become increasingly more precise and convincing, and it is likely that the latter published tetraploid genome will adopt the BioNano and Hi-C approaches (Hu et al., 2019; Wang et al., 2019) and improve the identification of homologous segments from subgenomes. The improved tetraploid cotton genome assemblies were consistent with FISH, which provides a reference for researchers deciding which genomes to adopt in their research.

Conclusions

We identified and characterized a new type of repetitive sequence termed ICRd motif in the Gossypium D genome. The motifs are interspersed in 13 chromosomes of the D genome, but absent in the A genome, and retained in D-subgenome in tetraploid cotton. We analyzed their structure, genomic distribution, affiliation, and evolution, which revealed a conserved region harbored in ancient LTR-TEs. The identification and characterization of the ICRd motif provided new insight into understanding TE evolution along with the formation of cotton genomes as well as providing a convenient and practical tool to distinguish the A and D genome subsets of the tetraploid cotton genome assembly. The ICRd motif has a novel structure and affiliation; how the structure was formed and what function the ICRd motif plays in LTR-TEs would be valuable areas for future research.

Supplemental Information

Table S1 Blast of the 1.8 kb sequences in G. raimondi genome

Click here for additional data file.

Table S2 The location of the ICRd motifs and their constituents in genome

Click here for additional data file.

Table S3 The structures of the LTR-TEs harboring the ICRd motif

Click here for additional data file.

Table S4 The information of the two homologous segments

Click here for additional data file.

Table S5 BLAST query with ICRd motif in tetraploid cotton genomes

Click here for additional data file.

Figure S1 The whole alignment of the 72 LTR-TEs

Click here for additional data file.

Doc S1 Two Perl scripts

Click here for additional data file.

Doc S2 The representative repetitive sequence

Click here for additional data file.

We are indebted to Dr Syed Shan-e-Ali Zaidi of the University of Liège, Belgium, for his guidance in analysis and interpretation of the data.

Additional Information and Declarations

Competing Interests

Author Contributions

Data Availability

The authors declare there are no competing interests.

Hejun Lu and Xinglei Cui analyzed the data, conceived and designed the experiments, performed the experiments, prepared figures and/or tables, authored or reviewed drafts of the paper, and approved the final draft.

Yanyan Zhao performed the experiments, prepared figures and/or tables, authored or reviewed drafts of the paper, and approved the final draft.

Richard Odongo Magwanga analyzed the data, performed the experiments, prepared figures and/or tables, authored or reviewed drafts of the paper, data analysis and interperetation, and approved the final draft.

Pengcheng Li performed the experiments, prepared figures and/or tables, data collection, and approved the final draft.

Xiaoyan Cai performed the experiments, and approved the final draft.

Zhongli Zhou, Xingxing Wang and Yuling Liu performed the experiments, authored or reviewed drafts of the paper, and approved the final draft.

Yanchao Xu and Yuqing Hou performed the experiments, prepared figures and/or tables, and approved the final draft.

Renhai Peng performed the experiments, authored or reviewed drafts of the paper, and approved the final draft.

Kunbo Wang and Fang Liu conceived and designed the experiments, authored or reviewed drafts of the paper, and approved the final draft.

The following information was supplied regarding data availability:

The sources of genome assemblies involved in this work: G. raimondii, https://www.cottongen.org/species/Gossypium_raimondii/jgi_genome_221; G. arboreum, https://www.cottongen.org/species/Gossypium_arboreum/bgi-A2_genome_v2.0; G. hirsutum, https://www.cottongen.org/species/Gossypium_hirsutum/bgi-AD1_genome_v1.0, https://www.cottongen.org/species/Gossypium_hirsutum/nbi-AD1_genome_v1.1, https://www.cottongen.org/species/Gossypium_hirsutum/HAU-AD1_genome_v1.0_v1.1, https://www.cottongen.org/species/Gossypium_hirsutum/ZJU-AD1_v2.1; G. barbadense, https://www.cottongen.org/species/Gossypium_barbadense/nbi-AD2_genome_v1.0, https://www.cottongen.org/species/Gossypium_barbadense/ZJU-AD2_v1.1, https://www.cottongen.org/species/Gossypium_barbadense/HAU-AD2_genome_v2.0.

The CAS version of G. barbadense is available in GenBank: PRJNA251673.

The Perl scripts are available as Supplementary File.

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
