# Peer review of "Identification of a genome-specific repetitive element in the Gossypium D genome"

_PeerJ, doi:10.7717/peerj.8344_

## Round 0.1 · original submission · Major Revisions

Aside of relatively minor edits, there is an important suggestion to re-run the analysis on new, improved genome assemblies. I understand that it creates a situation of a moving target (common in bioinformatics), but still, for the analysis of repeats this might indeed be essential. In addition, both reviewers insist on making scripts available.

Reviewer 1 ·

Basic reporting

No comments.

Experimental design

1 The Authors of the paper in the section «Material and methods» mention «in-house» Perl scripts, but do not provide links to the web resource where they can be downloaded. It is also necessary to provide a list of scripts and for each of them a brief description of what they are for.
2. Versions of genome sequences used in the work should be specified.
3. Versions of software used in the work should be specified.
4. It is not clear what the authors meant by the sentence «The insertions of repetitive sequences in genomes were illustrated by R language» in the 2.2 section of the paper. Did they mean that they wrote a script for this in R or something else?

Validity of the findings

No comments.

Reviewer 2 ·

Basic reporting

The manuscript is clear and concise, I have no questions on its context, aim and conclusions.
There are few minor points to change
e.g. lines 47-48 - "The genome-specific TE is an efficient approach to study species formation and genome evolution in genome comparative research" - looks like something ("investigation of" or "analysis of") is missing after "The".
line 97 - please correct reference to Ugene, this not Edgar 2004
line 120 - this should probably be "does not exist"
Regarding data sharing, the authors use custom scripts and accoding to PeerJ guidelines they should be made available, either as supplement or submitted to a public repository.

Experimental design

no comment

Validity of the findings

My major concern is that the authors use old version of genome assemblies of G. barbadense and G. hirsutum. In the recently published versions: https://www.nature.com/articles/s41588-019-0371-5 the assembly and annotation of repetitive regions is highly improved. I recommend to re-run the analysis concerning tetraploid cottons using these new assemblies.

---

## Round 0.2 · Minor Revisions

The new element should be compared to existing repeat databases.

Are there any other examples of subgenome-specific repeats or TEs? (Cytological works by the Paterson, Stelly, and Faridi groups have identified genome-specific repetitive elements.)

There are also typos mentioned by reviewer 2, and some remaining language issues (e.g. "repeat ... which repeats" in the Abstract).

Reviewer 1 ·

Basic reporting

I don't have any additional comments.

Experimental design

All issues mentioned in my previous review were corrected.

Validity of the findings

I don't have any additional comments.

Additional comments

I don't have any additional comments.

Reviewer 2 ·

Basic reporting

I found one typo in the species name: lane 90 - change G. arboretum to G. arboreum

Experimental design

no comments

Validity of the findings

no comments

Additional comments

All required revisions are made.

---

## Round 0.3 · Major Revisions

Dear authors,

I am the Section Editor for this part of the journal. We thank you for your contribution; however, as we are start to get to the refined manuscript there are some points that present some considerable concern which are not addressed.

Issue 1: There is no fully annotated sequence present within the text portion of the manuscript, except for the primers, and a representation of sequence files is not present in the supplemental data. A portion may appear in a figure, but is not accually considered as part of the text.

Issue 2: The CottonGen database is treated as the authority resource for repeat elements; this may not be the case. Is there a section in CottonGen that attributes itself an authority of the repetitive nature of the assembly? If not, then a database such as the GIRI Repbase database should be cited, and perhaps a common tool such a repeatmasker be used to provide a breakdown of the known elements known. The GIRI database resource represents sequences from several living organism resources; there may be others databases as well that focus of repetitive elements.

Issue 3: A reference sequence is used for select sequence data when it is commonly known that repetitive sequences are not assembled efficiently in the assembly biooinformatic process. A point of discussion would be needed to discuss what degree did they confirm that the genome assembly did not suffer from these consequences.

Issue 4: It would be imperative that either FISH work be included or that a measurement of the frequency of the sequences be confirmed, and that a true reference repetitive database be queried. There needs to be a clearer measurement of the assertions made in this manuscript to provide readers a baseline to either accept the finding or be able to validate them.

The manuscript misses these points and needs to be addressed. The manuscript is deemed to be placed in a major revision status until these points can be addressed.

---

## Round 0.4 · accepted · Accept

Thank you for your time considering critical comments referenced in previous reviews. It appears you have taken many of the comments to heart and addressed them as best as possible. The manuscript appears resolved in addressing concerns with explanation and illustration to describe the nature of this repeat element. The manuscript appears to be in good form and will be forwarded to be considered for publication. Please consider your manuscript accepted. Congratulations on your efforts.